# Effect of Polyphenols Intake on Obesity-Induced Maternal Programming

**DOI:** 10.3390/nu13072390

**Published:** 2021-07-13

**Authors:** Isabela Monique Fortunato, Tanila Wood dos Santos, Lucio Fábio Caldas Ferraz, Juliana Carvalho Santos, Marcelo Lima Ribeiro

**Affiliations:** 1Post Graduate Program in Health Science, Universidade São Francisco (USF), Bragança Paulista 12916-900, SP, Brazil; fortunato.misabela@gmail.com (I.M.F.); tanilawood@gmail.com (T.W.d.S.); lucio.ferraz@usf.edu.br (L.F.C.F.); 2Lymphoma Translational Group, Josep Carreras Leukemia Research Institute (IJC), 08916 Badalona, Spain

**Keywords:** bioactive compounds, nutraceutical, maternal obesity, adipose tissue, metabolic disorders

## Abstract

Excess caloric intake and body fat accumulation lead to obesity, a complex chronic disease that represents a significant public health problem due to the health-related risk factors. There is growing evidence showing that maternal obesity can program the offspring, which influences neonatal phenotype and predispose offspring to metabolic disorders such as obesity. This increased risk may also be epigenetically transmitted across generations. Thus, there is an imperative need to find effective reprogramming approaches in order to resume normal fetal development. Polyphenols are bioactive compounds found in vegetables and fruits that exert its anti-obesity effect through its powerful anti-oxidant and anti-inflammatory activities. Polyphenol supplementation has been proven to counteract the prejudicial effects of maternal obesity programming on progeny. Indeed, some polyphenols can cross the placenta and protect the fetal predisposition against obesity. The present review summarizes the effects of dietary polyphenols on obesity-induced maternal reprogramming as an offspring anti-obesity approach.

## 1. Introduction

Excess caloric intake and body fat accumulation lead to obesity. Obesity is considered as one of the main public health problems worldwide, and affects not only industrialized nations, but also developing countries. Epidemiological data presented by the World Health Organization have found that since 1975, obesity has almost tripled and at least 2.8 million people die annually due to complications from obesity or overweight. Despite the current public awareness of the consequences of obesity, its incidence continues to rise, and is distributed in almost all ethnicities and in both sexes, mainly affecting the population aged 25 to 44 years [1].

Obesity has a multifactorial etiology and involves an interaction between genetic and environmental factors [2]. It is characterized as white adipose tissue (WAT) expansion, and in general results from an energy imbalance as a result of an increase in caloric intake coupled with a decrease in daily energy spending [3,4]. The hyperplasia or hypertrophy of adipocytes caused by an accumulation of triacylglycerols leads to an expansion of body fat deposits and an increase in the concentrations of free circulating fatty acids, peptides, inflammatory cytokines, and adipokines, resulting in metabolic disorders such as hepatic steatosis, diabetes, metabolic syndrome, atherosclerosis, and dyslipidemia, conditions that also contribute to the cardiometabolic risk [5,6].

Maternal obesity also has different clinical implications, and is associated with significant health risks to the mother and the newborn. Of note, in animal models, high fat diet (HFD) intake during the pregnancy may predispose offspring to postnatal metabolic inflammatory disorders such as obesity. In this sense, the well-known anti-inflammatory activity of polyphenols could be considered as a reprogramming strategy against maternal obesity-induced adversities. Thus, the present review summarizes the effects of dietary polyphenols on obesity-induced maternal reprogramming as an offspring anti-obesity approach.

## 2. Obesity and Maternal Programming

It has been described that nutritional, hormonal, and environmental changes during pregnancy and lactation are strongly associated with the appearance of adulthood diseases. The physiological adaptations developed by the organism, as a survival strategy, could potentially become detrimental to the individual’s health from the moment when adverse conditions are restored to normal levels. Although several studies have shown that these disorders can have origins even before birth, the exact mechanisms by which these alterations occur are still poorly understood. This biological phenomenon was initially called “metabolic programming” [7,8,9] and later as “ontogenetic plasticity”, as it is a more probabilistic than a deterministic event [10].

Indeed, the phenomenon of programming or ontogenetic plasticity has been extensively studied for decades by several research groups around the world. Adequate nutrition is known to be essential during crucial periods of development as it can act as an imprinting or priming factor, leading to physiological changes programming future diseases. The association between the observed changes in critical periods and disorders in adulthood gave rise to the “Barker hypothesis”, then the Fetal Origin of Adult Disease, and is now the Developmental Origins of Health and Disease (DOHaD) [11].

Several factors involved in disease-programming have already been described [12,13,14,15]. Among them, we highlight maternal obesity, which has been widely linked to the birth of newborns who are more susceptible to overweight and obesity development [16,17]. Although the prevalence of obesity grows alarmingly and affects all age groups, it should be noted that approximately 39.7% of women between 20 and 39 years of age are obese in the U.S. (NCHS Data Brief No. 360, February 2020). This dataset shows that obesity is a seriously concerning vicious cycle that requires intensive research.

Maternal obesity has been associated with abdominal fat expansion and higher cardiometabolic risk (increased blood pressure, increased serum concentrations of triglycerides, LDL-C and C-Reactive Protein, and low levels of HDL-C) [18]. In animal studies, maternal obesity is correlated with increased adiposity, hyperphagia, increased cholesterol and triglyceride concentrations and increased lipogenesis and a decrease in beta oxidation, leading to the development of hepatic steatosis on offspring [15,19]. Furthermore, it has already been demonstrated that the progeny of HFD-fed rats showed changes such as weight gain, increased adiposity, hyperleptinemia, leptin resistance, and hyperglycemia, even immediately after weaning (21 days) [20].

The deleterious consequence of maternal obesity occurs during specific periods of fetal and offspring development, where epigenetic memory can be switched [21]. Murine models have shown that adipogenesis is active, especially during the last week of gestation, and accelerates in early postnatal life until the puppies are weaned. In humans, adipogenesis and adipose tissue growth occur mainly before birth [9]. In both, the expansion of adipose tissue involves hyperplasia and hypertrophy. In adult mice, the adipocyte reserve remains quite stable, ranging from 10 to 20% of adipocyte renewal per month [22]. The number of fat cells in humans is established in adulthood, with a renewal of 10% per year, regardless of body weight, which means that obese persons renew twice as many fat cells annually. Furthermore, when obese persons lose a substantial amount of weight, they maintain their high number of adipocytes, indicating that the adipose tissue has a numerical adipocyte memory, which is defined at the beginning of development [9]. Thus, it is suggested that multiple epigenetic factors are involved in “memorizing” the number of adipocytes.

## 3. Adipose Tissue Programming

Human adipose tissue development is an uninterrupted process that starts early during embryogenesis [23]. The transcriptional cascade promoting adipogenesis is very complex and has been well-studied. Mechanically, during the adipose tissue development, the stem cells are sensitive to pro-adipogenic signals (such as metabolites and hormones). These signals stimulate several epigenomic changes in transcriptionally accessible regions of adipogenic genes. The epigenomic profiles established during adipogenesis take place in two steps: (1) differentiation of multipotent mesenchymal stem cells (MSCs) into preadipocytes, and (2) terminal adipocyte differentiation [24].

Stem cells have active transcription of pluripotency genes, and repression of adipogenic genes through the presence of epigenetic marks such as DNA methylation (H3K27me3 and H3K9me3) [9,24]. During the pre-adipogenic process, pro-adipogenic genes are silenced by the presence of bivalent histones H3K27me3 and H3K4me3 [25]. The bivalent state is converted to an active state during the adipocyte terminal differentiation, which is associated with the recruitment of the first wave of transcription factors (such as CCAAT Enhancer Binding Protein (C/EBP)β, glucocorticoid receptor, Sterol Regulatory Element Binding Transcription Factor 1 (SREBP-1C), and Zinc Finger Protein 423 (ZFP423)) [25,26,27,28]. These transcription factors, in turn, stimulate the open state of chromatin at regions containing genes involved in the second wave of transcription factors, among which Peroxisome Proliferator Activated Receptor (PPAR)γ and CEBPα stand out. The second wave of transcription factors leads to the transcription of pro-adipogenic genes (such as Solute Carrier Family 2 Member 4 (SLC2A4), Adiponectin (ADIPOQ), and Leptin (LEP)) [25]. The pregnancy and lactation period involve a strong adipogenesis commitment. The adipose tissue, in turn, originates in both embryonic and postnatal development, especially during early childhood and puberty. In fact, in rodents and in humans, there is a gradual gain in adipocyte number until puberty under physiological conditions [29,30]. The number and size of adipocytes remain stable after puberty [29,31].

Several molecular evidences suggest that transmission of epigenetic marks such as DNA methylation and changes in histone code may be important in the children’s phenotypes. Recent studies suggest that the emergence of chronic diseases such as obesity and diabetes may have an epigenetic inheritance [32,33,34]. Accordingly, it has been shown in rodent models that maternal obesity modulates the expression of pro-adipogenic genes such as C/EBPβ, ZFP423, and PPARγ in the offspring, thus reprogramming adipogenesis. In a maternal obesity-programming model, it was observed that the offspring exhibited an increase in the levels of ZFP423 expression, which was associated with a reduced methylation in its promoter. At weaning, it was observed that the high activity ZFP423 led to faster adipogenesis, increased adipocyte differentiation, and higher adiposity [35,36,37]. In addition, DNA methylation and histone modifications may regulate PPARγ expression in adipose tissue. Thus, maternal obesity inhibits PPARγ2 expression through epigenetic mechanisms in the WAT of the offspring. Although it seems paradoxical, it is believed that the repression of this gene in adipose tissue may be an adaptive mechanism to avoid further accumulation of fat [38,39]. However, the persistent repression of PPARγ2 is associated with an increase in the expression of genes involved with fat accumulation (TNF receptor superfamily member 6 (Fas), diacylglycerol O-acyltransferase 2 (Dgat2), and lipoprotein lipase (Lpl)), suggesting that, at least in part, such regulation should occur by additional signaling pathways independent of PPARγ [15]. In summary, maternal obesity may be related to important epigenetic modifications that explain the emergence of obesity and other morbidities presented by the offspring.

## 4. Polyphenol Effect on Obesity

Beneficial effects of some natural diet components are attributed to their anti-inflammatory and anti-oxidant properties, and have been commonly used to treat and/or prevent diseases. In this sense, several studies highlight a broad range of beneficial effects to polyphenols [40,41].

Polyphenols are bioactive compounds abundantly found in vegetables and fruits involved in plant defense against oxidative stress and ultraviolet radiation, or attracting pollinators [42]. Since chronic inflammation is known to be a leading cause of different disorders such as obesity, diabetes type 2, cancer, arthritis, neurodegenerative, and cardiovascular diseases, many studies have studied the beneficial role of polyphenols in reducing inflammation to treat chronic disorders [43].

These natural compounds contribute to the regulation of inflammatory signaling through the modulation of several pro-inflammatory genes such as cytokines, lipoxygenases, nitric oxide synthases cyclooxygenase, and through their anti-oxidant activity, which support progress toward decreased metabolic disorders [44,45].

Polyphenols belong to a broad group of chemical metabolites, and can be categorized into flavonoids, allied phenolic, and polyphenolic compounds [46]. Some polyphenols such as anthocyanins are absorbed through the gut barrier, while the unabsorbed polyphenols must be enzymatically hydrolyzed to be uptaken by epithelial cells [47]. The anti-obesity effect ascribed to polyphenols are achieved by their ability to interact, directly or indirectly, with adipose tissues and activate 5′ adenosine monophosphate-activated protein kinase (AMPK), which results in the reduction of cholesterol, fatty acid synthesis, and triglyceride formation. Moreover, polyphenols can repress genes that regulate adipocyte differentiation and triglyceride accumulation [48].

Epigallocatechin gallate (EGCG) is the most active flavonoid compound present in green tea. Strong emerging evidence has shown the anti-obesity potential of EGCG. For instance, in vitro studies have demonstrated that EGCG inhibits preadipocyte differentiation, decreases adipocyte proliferation, suppresses lipogenesis, induces adipocyte apoptosis, and promotes lipolysis and fatty acid β-oxidation [49,50,51]. Additionally, EGCG decreased obesity and epididymal WAT weight in mice partially via activating the AMPK pathway [52]. Adipocyte differentiation may be suppressed by EGCG through the inhibition of the PI3K/AKT and MEK/ERK pathways, which may lead to downregulation of PPARγ and C/EBPα, the main adipogenesis regulators [48].

Quercetin is one of the primary flavonoid compounds and widely exists in vegetables, tea, and fruits. It has been reported that quercetin suppresses lipid accumulation, body weight, and insulin resistance in mice [53]. In HFD-induced rodents, treatment with quercetin attenuated both obesity and insulin resistance, inhibited hepatic lipid accumulation by inducing the expression of beta-oxidation related genes and decreasing inflammation [54,55]. In human adipocyte models, quercetin significantly downregulated adipokines (i.e., Angiopoietin Like 4 (ANGPTL4) and Serpin Family E Member 1(SERPINE1, previously known as PAI-1)) and glycolysis-associated enzymes (i.e., Enolase 2, gamma neuronal (ENO2) and 6-phosphofructo-2-kinase/fructose-2,6-biphosphatase 4 (PFKFB4)), which are closely related to obesity [56]. Mechanistically, in vitro and in vivo analysis has shown that quercetin is able to revert unfavorable epigenomic profiles associated with adipogenesis through the induction of chromatin remodeling and histone modifications, which lead to a decrease in C/EBPα and PPARγ gene expression [55].

In a HFD-fed rat model, kaempferol suppressed the visceral fat accumulation and improved hyperlipidemia through the downregulation of SREBPs and upregulation of hepatic PPARα expression [57]. In obese HFD-induced mice, kaempferol protected against obesity and ameliorated hyperlipidemia partly through maintaining microbial diversity and modulating microbial communities as well as downregulating PPARγ and SREBP-1C [58,59]. In another study, kaempferol modulated 3T3-L1 adipocyte differentiation through regulation of C/EBPα and PPARα [60,61].

Curcumin, the primary natural polyphenolic compound of the spice turmeric, also has anti-insulin sensitivity and anti-obesity activity. It has been shown that curcumin can suppress weight gain, improve insulin sensitivity, and prevent the development of diabetes in rodents and in prediabetic subjects [62]. Curcumin promoted beige adipogenesis and induced pre-adipocyte apoptosis in white adipocytes possibly mediated by AMPK, PPARγ, and C/EBPα [63,64]. In pro-adipogenic conditions of rat MSCs, curcumin inhibited the expression of PPARγ and C/EBPα, which in turn suppressed adipocyte differentiation [65].

Resveratrol, one of the most studied bioactive polyphenolic compounds found in grapes, red wine, and some berries, also modulates several events involved in the obesity process. Due its powerful antioxidant and anti-inflammatory activity, resveratrol has been used as dietary supplements, and as a functional food ingredient to achieve different health benefits. Several in vitro studies have demonstrated that resveratrol has an anti-obesity effect by negatively regulating white adipogenesis via inhibiting PPARγ and C/EBPα [66], activating Sirtuin 1 (SIRT1) and (PPARγ Coactivator 1) PGC1α [67], attenuating white adipogenesis and lipid accumulation. Furthermore, their anti-obesity effect is carried out by inhibiting the transcriptional activities of PPARγ, suppressing adipocyte differentiation, proliferation, and lipogenesis, and promoting adipocyte apoptosis, lipolysis, and fatty acid oxidation [68,69,70].

Several studies have shown that chlorogenic acid (CGA), the major polyphenol present in the coffee, has beneficial effects on obesity. It has been demonstrated, in an HFD-induced mice obese model, that CGA supplementation significantly decreased body weight, visceral fat mass, and leptin and insulin plasma levels [71,72]. Mechanistically, the CGA beneficial anti-obesity effects might be attributed to its ability to decrease C/EBP, PPARγ, and SREBP expression [72]. Furthermore, an interesting study conducted in mice that become identifiably obese due to homozygous diabetes spontaneous mutation (Lepr^db^) showed that CGA treatment inhibited G6Pase expression, improved lipid metabolism, insulin sensitivity, and glucose tolerance via AMPK activation [73].

Increasing clinical evidence has also shown that coffee exerts anti-obesity effects in humans [74]. Nordestgaard et al. [75] conducted a Mendelian randomization study including 93,179 subjects, and showed that the consumption of up to four cups of coffee per day was associated with a lower risk of obesity than non-coffee drinkers. Similarly, Koyama et al. [76] reported that the daily coffee intake was associated with lower levels of visceral obesity and metabolic syndrome in a population of 3539 Japanese. Another clinical study indicated that 30 obese women who consumed 180 mg of CGA for eight weeks presented significant reductions in body weight, BMI, and fat mass indices as well as lower serum LDL, TC, leptin, and plasma free FA levels when compared with 34 obese women from the placebo group [77].

Salvianolic acid A (SA) is another natural polyphenolic compound found in *Radix Salvia miltiorrhiza*, with well reported anti-oxidant and anti-inflammatory properties [78]. It has been shown that SA can also reverse the HFD-induced obesity [79]. Indeed, a study conducted with HFD-induced obese mice showed that SA supplementation exerts its anti-obesity effects by activation of WAT browning via the AMPK-SIRT1 pathway [80]. Moreover, SA has a cardioprotective role against lipotoxicity by suppressing the expression of TLR4 target genes, both in vitro and in vivo [81].

Ferulic acid (FA), a polyphenol abundantly found in whole grains, also has the ability to activate AMPK signaling and exerts its anti HFD-induced obesity effects by inhibiting oxidative stress, inflammation, and circulating LDL levels along with increasing adiponectin expression and circulating HDL levels [82]. Additionally, using a 3T3-L1 pre-adipocytes in vitro model, Kuppusamy et al. [83] demonstrated that FA may prevent obesity through downregulation of key transcriptional factors PPARγ and C/EBPα and consequently suppressing adipocyte differentiation and lipid accumulation.

Recent data have shown that TOTUM-63 (T63), a novel plant-based polyphenol extracted from olive leaves, bilberry, artichoke, chrysanthellum, and black pepper, may also have anti-obesity properties. Indeed, a translational study indicated that T63 can reduce body weight, control glucose homeostasis, and may protect against type 2 diabetes [84]. Accordantly, recent studies based on obese rodent models have shown that T63 supplementation improves glycemic levels by inducing skeletal muscle oxidative capacity [85], decreases inflammation, and improves insulin sensitivity [86], suggesting that T63 might be a promising novel nutritional supplement against obesity. Figure 1 summarizes the molecular mechanisms underlying the anti-obesity effects of the main polyphenols.

## 5. Beneficial Impact of Polyphenols on Obesity-Induced Maternal Programming

It has been shown that maternal obesity functionally modulates adipogenesis from fetal development [87] to adulthood [88], presenting larger adipocytes and WAT mass accumulation [89]. Moreover, maternal obesity leads to increased cytokine production and placental-mediated inflammation, which could affect fetal development and may predispose progeny to subsequent obesity [90,91]. Since maternal obesity-induced metabolic programming has a profound impact on offspring, there is a compelling need to find effective reprogramming approaches in order to resume normal development. In this sense, supplementing with natural compounds such as polyphenols could be helpful in the reprogramming of maternal adversities associated with obesity.

A study conducted by Kataoka, Norikura, and Sato (2018) [92] showed that the intake of EGCG–rich green tea extract during lactation of Wistar rats had a protective effect on the kidney of HFD-fed adult offspring through the suppression of epigenetic modulators such as DNA methyltransferase 1 (DNMT1), ubiquitin like with PHD and ring finger domains 1 (UHRF1), and euchromatic histone lysine methyltransferase 2 (EHMT2), highlighting that the offspring phenotype can be programmed by maternal polyphenol intake. In addition, EGCG treatment of rat embryos blocks Forkhead Box O3 (FOXO3A) activation and reverses AKT inhibition, preventing hyperglycemia-induced embryopathy [93]. Similarly, it has been shown that an isoflavone-rich soy-based diet during rat pregnancy may afford cardiovascular protection in the offspring through the stimulation of antioxidant and redox-sensitive gene expression [94].

Since efficiency and safety of EGCG supplementation has rarely been evaluated during pregnancy in humans, a well-conducted randomized, placebo-controlled, double-blind clinical study evaluated the effects of EGCG in gestational diabetes mellitus-affected women. In general, the authors described that daily administration of 500 mg of EGCG, starting from the beginning of their third trimester, was well tolerated and improved both circulating glucose and insulin response. Additionally, EGCG also ameliorates some neonatal complications such as low birth weight and hypoglycemia [95].

Interestingly, Tain Y et al. [96] highlighted the beneficial effects of resveratrol supplementation as an important reprogramming approach against the metabolic syndrome-related disorders. The long-term effects of both maternal and postnatal HFD intake lead to metabolic disruption characterized by body weight gain, high levels of serum HDL, ALT, triglycerides, leptin, cholesterol, and angiotensin I and II, which could be ameliorated by resveratrol therapy [97]. It has been suggested that resveratrol could cross the placenta and exert its anti-inflammatory [98] and anti-oxidant function [99]. Animal studies have shown the fat browning activator action of resveratrol involving the secretion of several myokines and adipokines and suggest that maternal resveratrol supplementation may have a protective role against HFD-induced obesity [100,101]. Similarly, Zou et al. found that the resveratrol supplementation during pregnancy led to a thermogenic program in BAT and WAT, and stimulated beige adipocyte development of WAT in male progeny [102]. Hsu et al. [103] assessed the effect of maternal resveratrol intake of obese rats and found that resveratrol treatment restored adiponectin, AKT phosphorylation, and brain-derived neurotrophic factor (BDNF) in male fetal brain, increased blood pressure, and reduced increased body weight and peripheral insulin resistance in adult male offspring, further demonstrating that intervention with this polyphenol may protect offspring against HFD-induced obesity. Another study using the HFD-obese rat model [104] highlighted that maternal resveratrol supplementation had a reprogramming role for progeny through lipid metabolic modulation. The authors showed the anti-obesity effect of resveratrol treatment through the suppression of lipogenesis and SIRT1 protein expression, attenuation of leptin resistance, and induction of lipolysis for offspring. Additionally, it has been shown that resveratrol supplementation prevents maternal glucose intolerance and lower blood glucose levels by insulin secretion stimulation [105]. Figure 2 summarizes obesity-induced maternal programming and how polyphenol intake could reprogram the epigenetic memory of adult offspring.

Although there is growing evidence suggesting the therapeutic potential of resveratrol in obesity-induced maternal reprogramming, the safety and long-term risk of in utero exposure in humans are unexplored. Thus, the effects of resveratrol supplementation was evaluated in a Western-style diet-fed pregnant nonhuman primates [106]. In this interesting study, the authors showed that resveratrol supplementation in pregnancy has beneficial effects such as a reduction in maternal weight, an improvement on insulin secretion, restore uteroplacental blood flow, decreased placental inflammation, and an improvement in lipid deposition in the fetal liver. However, an increase in fetal pancreas mass was observed. Altogether, these data highlight that the use of resveratrol should be carefully evaluated in pregnant women.

## 6. Conclusions

The current evidence highlights the relevance of maternal obesity for the metabolic health of the progeny. This review shows that there is an increasing amount of experimental data pointing to the potential effects of polyphenols as a strategy to counteract the deleterious effects induced by maternal obesity. In general, the data reviewed here demonstrated that supplementing pregnant and lactating obese animals with polyphenols including resveratrol, genistein, EGCG, and anthocyanins led to metabolic health reprogramming that ultimately decreased adiposity in the offspring. Whether these observations translate to the human condition remains to be determined. Further examination of obesity-induced maternal programming, especially in humans, is urgently needed and may help to develop polyphenol-based strategies to decrease the propagation of obesity across generations.

## Figures and Tables

**Figure 1 nutrients-13-02390-f001:**
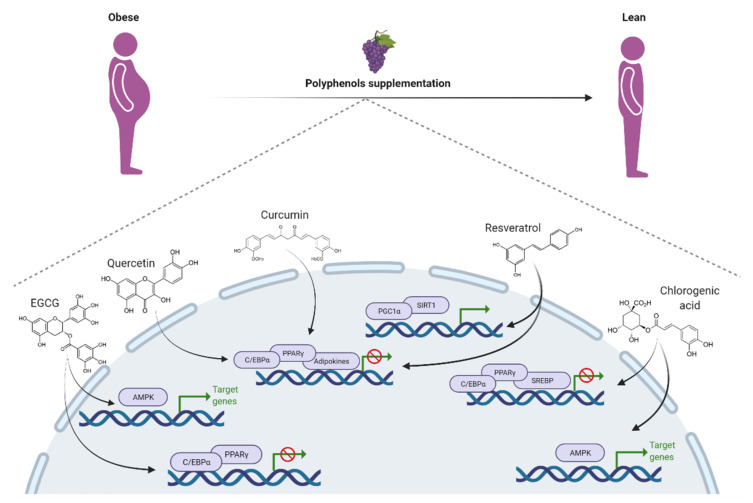
Schematic representation of molecular mechanisms underlying the anti-obesity effects of the main polyphenols.

**Figure 2 nutrients-13-02390-f002:**
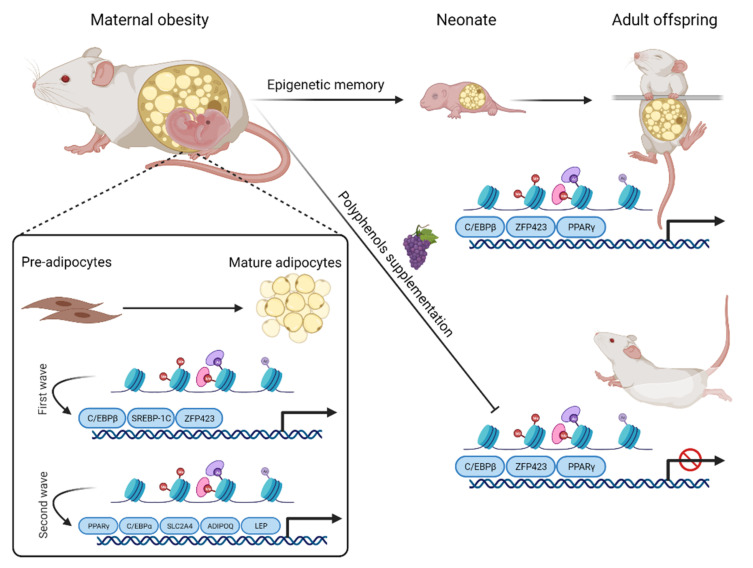
Schematic representation of polyphenol effect on obesity-induced maternal programming. The recruitment of the first wave of transcription factors (C/EBPβ, SREBP-1C, and ZFP423) take place during adipocyte differentiation, leading to the conversion of the histone bivalent state to an active state. These transcription factors, in turn, promote an open state of chromatin in regions containing genes involved in the second wave of transcription factors, among which PPARγ and CEBPα stand out. The second wave of transcription factors induces the expression of pro-adipogenic genes such as LEP, SLC2A4, and ADIPOQ. Maternal obesity is also involved in the regulation of pro-adipogenic transcription factors such as ZFP423, C/EBPβ, and PPARγ during adipogenesis in the perinatal period and affects the offspring. Polyphenol supplementation could counteract the detrimental effects of maternal obesity programming on the progeny by negatively regulating adipogenesis via inhibiting PPARγ, ZFP423, and C/EBPα.

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
