# Peer review of "Effect of Polyphenols Intake on Obesity-Induced Maternal Programming"

_nutrients, 2021, doi:10.3390/nu13072390_

Round 1
Reviewer 1 Report
This review article begins with a nice clear and succinct abstract and a good introduction to the important topic of obesity, and the impact on maternal health and offspring development. The structure of the review content is well laid out and walks the reader through the relevant information. There are some edits that will make this review article more appropriate for publication:
For clarity, I would suggest that the content of section 4 (current heading 'Polyphenols effect on obesity and maternal programming') be divided in to two parts. The first should address studies of polyphenols and the mechanisms of action in non-pregnant studies (as it currently does). At line 229, polyphenols in pregnancy are discussed. This should be under a separate heading to distinguish pregnancy studies.
I have one concern with the literature that has been cited. The review has focused on rodent studies of polyphenols and not considered larger animal models. In general this may not need to be extensively considered in this review but important data cannot be overlooked. In particular, as the authors have focused their attention on resveratrol, the study by Roberts et al., (2014) conducted in nonhuman primates should be included. This study demonstrated beneficial effects of resveratrol supplementation in high fat diet pregnancies but cautioned the unexplained developmental effect on the fetal pancreas. In a review intended to summarize the literature for the reader, a comprehensive perspective must be provided to correctly report any potential detrimental impact of polyphenols use in pregnancy.
Author Response
This review article begins with a nice clear and succinct abstract and a good introduction to the important topic of obesity, and the impact on maternal health and offspring development. The structure of the review content is well laid out and walks the reader through the relevant information. There are some edits that will make this review article more appropriate for publication:
Comment 1 - For clarity, I would suggest that the content of section 4 (current heading 'Polyphenols effect on obesity and maternal programming') be divided in to two parts. The first should address studies of polyphenols and the mechanisms of action in non-pregnant studies (as it currently does). At line 229, polyphenols in pregnancy are discussed. This should be under a separate heading to distinguish pregnancy studies.
Answer - We are grateful for the reviewer’s comment. Following his/her recommendation, section 4 be divided into two parts.
Comment 2 - I have one concern with the literature that has been cited. The review has focused on rodent studies of polyphenols and not considered larger animal models. In general this may not need to be extensively considered in this review but important data cannot be overlooked. In particular, as the authors have focused their attention on resveratrol, the study by Roberts et al., (2014) conducted in nonhuman primates should be included. This study demonstrated beneficial effects of resveratrol supplementation in high fat diet pregnancies but cautioned the unexplained developmental effect on the fetal pancreas. In a review intended to summarize the literature for the reader, a comprehensive perspective must be provided to correctly report any potential detrimental impact of polyphenols use in pregnancy.
Answer - We are grateful for the reviewer’s comment. Following his/her recommendation, the excellent manuscript Robert et al., (2014) was added (Lines 345-354). In addition, we also added the efficiency and safety of EGCG supplementation in gestational diabetes mellitus-affected women (Lines 312-318).
Reviewer 2 Report
This manuscript is a mini-review which described the effect of polyphenols on obesity. The review clearly introduces obesity issue in the current time and some causes in adipose tissue programming to occur the disease. However, authors described only four polyphenol compounds in the section 4 part. the limited number of the polyphenols and the researches were shown in the section. Considering a review manuscript, it should survey and summary for various polyphenol compounds effect on the anti-obesity or recently research results as tables or figures with chemical structures. In current statue, the data in this review are not enough to explain the correlation of polyphenols' positive effects on the obesity.
Author Response
This manuscript is a mini-review which described the effect of polyphenols on obesity. The review clearly introduces obesity issue in the current time and some causes in adipose tissue programming to occur the disease.
Comment 1 - However, authors described only four polyphenol compounds in the section 4 part. the limited number of the polyphenols and the researches were shown in the section. Considering a review manuscript, it should survey and summary for various polyphenol compounds effect on the anti-obesity or recently research results as tables or figures with chemical structures. In current statue, the data in this review are not enough to explain the correlation of polyphenols' positive effects on obesity.
Answer - We are grateful for the reviewer’s comment. Following his/her recommendation, we changed substantially section 4 adding the effects of other polyphenols on obesity (such as chlorogenic acid, salvianolic acid A, ferulic acid, and TOTUM-63, Lines 222-264). We hope that these changes are sufficient to emphasise the important role of polyphenols in obesity. We also add a new figure (Figure 1, Page 6), highlighting the molecular mechanisms underlying the anti-obesity effects of some polyphenols.
Reviewer 3 Report
Obesity is considered as one of the main public health problems in the world. Over the last years, nutraceutical has aroused considerable interest for the beneficial effects that some nutrients (including polyphenols) contained in food have on health, on the prevention and the treatment of illnesses.
The topic on polyphenols and obesity falls in the scope of this journal. The design is reasonable and the authors have given good number of citations about the subject. I have read the manuscript with interest, I found it very complete, detailed and well written. Therefore, I believe it is suitable for publication.
However, I recommend some small changes, as described below.
Line 112: replace “are” with “is”;
Line 133: replace “children phenotypes” with “children’s phenotypes”;
Line 239: replace “a protective effects” with “a protective effect”.
I suggest to change the keywords because it is better to avoid repetition of words already present in the title. By using words not present in the title, you increase the chance of finding it in articles searches. For example, you could replace with “bioactive compounds, nutraceutical, maternal obesity, adipose tissue, metabolic disorders”.
Author Response
Obesity is considered as one of the main public health problems in the world. Over the last years, nutraceutical has aroused considerable interest for the beneficial effects that some nutrients (including polyphenols) contained in food have on health, on the prevention and the treatment of illnesses.
The topic on polyphenols and obesity falls in the scope of this journal. The design is reasonable and the authors have given good number of citations about the subject. I have read the manuscript with interest, I found it very complete, detailed and well written. Therefore, I believe it is suitable for publication.
Comment 1 - However, I recommend some small changes, as described below.
Line 112: replace “are” with “is”;
Line 133: replace “children phenotypes” with “children’s phenotypes”;
Line 239: replace “a protective effects” with “a protective effect”.
Answer - We are grateful for the reviewer’s comment. Following his/her recommendation, all changes were done
Comment 2 - I suggest changing the keywords because it is better to avoid the repetition of words already present in the title. By using words not present in the title, you increase the chance of finding it in articles searches. For example, you could replace it with “bioactive compounds, nutraceutical, maternal obesity, adipose tissue, metabolic disorders”.
Answer - We are grateful for the reviewer’s comment. Following his/her recommendation, we changed the keywords.
Round 2
Reviewer 2 Report
The reviewers' comments were well revised.